# Personalized Brachytherapy: Applications and Future Directions

**DOI:** 10.3390/cancers16193424

**Published:** 2024-10-09

**Authors:** Piyush Pathak, Justin J. Thomas, Arjit Baghwala, Chengfeng Li, Bin S. Teh, Edward B. Butler, Andrew M. Farach

**Affiliations:** 1Department of Radiation Oncology, Baylor College of Medicine, Houston, TX 77030, USA; justinjthomas001@gmail.com; 2Department of Radiation Oncology, Houston Methodist Hospital, Houston, TX 77030, USA; abaghwala@houstonmethodist.org (A.B.); cli3@houstonmethodist.org (C.L.); bteh@houstonmethodist.org (B.S.T.); ebutler@houstonmethodist.org (E.B.B.)

**Keywords:** personalized brachytherapy, image-guided adaptive radiotherapy, 3D-printed applicators, intensity-modulated brachytherapy, shielded brachytherapy, genomic biomarkers, artificial intelligence in radiation oncology, brachytherapy–immunotherapy combination, treatment planning optimization, precision oncology

## Abstract

**Simple Summary:**

Brachytherapy is a form of internal radiation therapy where radioactive sources are placed directly in or near tumors. This paper shows how brachytherapy can be personalized using new technologies like 3D-printed applicators, advanced imaging techniques, and artificial intelligence to make treatment more precise and effective. The authors also explore the role of genetic tests and biomarkers for choosing the best treatments for each patient, as well as future approaches such as combining brachytherapy with immunotherapy and developing new ways to shape radiation doses using shielding. By tailoring treatments to individual patients, personalized brachytherapy aims to effectively treat cancer while reducing treatment-related side effects.

**Abstract:**

Brachytherapy offers a highly conformal and adaptive approach to radiation therapy for various oncologic conditions. This review explores the rationale, applications, technological advances, and future directions of personalized brachytherapy. Integration of advanced imaging techniques, 3D-printed applicators, and artificial intelligence are rapidly enhancing brachytherapy delivery and efficiency, while genomic tests and molecular biomarkers are refining patient and dose selection. Emerging research on combining brachytherapy with immunotherapy offers unique synergistic potential, and technologies such as intensity-modulated and shielded brachytherapy applicators present novel opportunities to further optimize dose distributions. Despite these promising advances, the field faces challenges including a need to train more practitioners and develop new approaches to treating a broader range of malignancies. As personalized medicine evolves, brachytherapy’s ability to deliver highly targeted, individualized treatments positions it as a critical component in future cancer care.

## 1. Introduction

Brachytherapy (BT) is widely recognized as the most ablative, conformal, and tissue-sparing radiotherapy (RT) modality. Given its procedural nature, however, outcomes are dependent on both practitioner skill and access to a variety of techniques and applicators. The dwindling utilization of BT has resulted in decreased training opportunities in advanced BT practice, compounding the problem of widespread generalizability and standardization [1,2]. However, the greatest purported limitation of BT, a lack of standardization, can also serve as the modalities’ most valued asset: absolute personalization.

### Rationale for Personalized Brachytherapy

The goal of RT is to deliver tumoricidal doses while minimizing dose to organs at risk (OARs) and resultant toxicity. Brachytherapy inherently offers several advantages that make it ideal for optimizing these objectives in a patient- and tumor-specific manner:Proximity to tumor: BT allows for precise placement of radiation sources directly within or near the tumor, enabling highly conformal dose delivery.Steep dose gradient: A typical dose gradient of 5–20% per millimeter at the target volume periphery for a 192Ir source allows for rapid dose fall-off, sparing nearby OARs [3].Real-time visualization: Image guidance enables visualization of targets and OARs in real time, allowing for plan optimization based on current anatomy rather than pre-treatment imaging [4].Shorter treatment course: BT is typically delivered over a shorter duration than external beam radiotherapy, reducing intra-fraction motion considerations.Inter-fraction responsiveness: BT’s ability to adapt to tumor response and anatomical changes between fractions ensures appropriate target coverage and limits OAR dose.

Personalizing brachytherapy leverages these inherent advantages, leading to improved local control, reduced toxicity, and better overall outcomes for patients across various cancer types.

## 2. Equipment Customization

The selection of the optimal BT applicator is based on several factors, including the type of cancer, the tumor’s location, and the patient’s anatomy. Although there are standard applicators available for routine use (e.g., vaginal cylinders), customized applicators are becoming more widespread, with a recent emphasis on 3D printing. This technology has enabled individual clinics to craft highly personalized, single-use templates and applicators while keeping costs manageable and mitigating the need for specialized BT training by prefabricating complex conformal shapes and catheter paths. This is particularly beneficial for interstitial BT, where accurate needle placement is crucial for optimal dosimetry.

A recent systematic review highlights the 3D printing workflow and notes accelerating utilization over time, with gynecological applications representing the highest percentage (32%), followed by skin (19%), and head and neck (9%) [5]. Three main areas of clinical use have been explored: applicators, templates, and quality assurance (QA) devices, with the focus of studies evolving over time. Early research primarily concentrated on the feasibility of using 3D printing for BT applicators and the evaluation of materials suitable for clinical use. More recent studies have shifted towards clinical implementation, dosimetric comparisons, and the development of novel designs, including shielded applicators for intensity-modulated brachytherapy (IMBT) [5,6]. Several studies have shown an overall favorable impact of 3D-printed customized applicators on dosimetric outcomes, repeatability, and procedure time using both low-dose rate (LDR) and high-dose rate (HDR) modalities, including Yuan et al.’s prospective randomized series [5,7,8,9].

While 3D printing has shown rapid clinical utilization, others have shown a variety of methods that accomplish similar customization potential, including acrylic resin-fabricated applicators [10] and silicone bolus creation for uncommon BT applications such as penile cancer [11]. Below, we present two cases demonstrating the wide clinical potential of BT with readily available materials. In the first, we show a unique implementation of custom silicone mold applicators for skin cancer of the thumb; the second demonstrates how an existing gynecologic skin template can be repurposed for salvage therapy of rectal cancer.

### 2.1. Patient 1

*Clinical Scenario and Indications:* A 71-year-old male presented with squamous cell carcinoma in situ of the thumb refractory to topical therapy (Figure 1A) and history of immunosuppression due to prior cardiac transplant. Digital BT is not common due to the precision required to treat such a small and anatomically complex area without compromising function.

*Brachytherapy Technique and Applicators:* The patient was treated with surface mold BT, customized to encircle the involved portion of the thumb and nail bed while sparing the uninvolved portion of the digit to minimize the risk of lymphedema (Figure 1B). A total of 4000 cGy was delivered in 8 fractions, twice weekly.

*Outcomes:* The patient tolerated the treatment well with expected grade 1 dermatitis. Using this technique, durable local control rates exceeding 95% [12] and preserved long-term digit function are anticipated. This demonstrates the effectiveness of digital BT in challenging cases.

### 2.2. Patient 2

*Clinical Scenario and Indications:* A 50-year-old male presented with local recurrence of rectal cancer after prior neoadjuvant chemoradiation and surgery. Rectal BT is rarely performed due to the complexities involved in accessing the tumor site and the potential risks but was considered necessary for this salvage treatment.

*Brachytherapy Technique and Applicators:* HDR BT was conducted over five sessions, utilizing a transrectal ultrasound (TRUS) for precise needle placement and gold fiducial markers for improved target visualization (Figure 2A). A Kelowna gynecologic template (Figure 2B; Varian Medical Systems, Palo Alto, CA, USA) was used to stabilize 7 to 11 transperineal needles (lower needle counts were utilized in later fractions due to tumor shrinkage), delivering a total dose of 2500 cGy.

*Outcomes:* The patient tolerated treatment well, with an excellent clinical response and minimal acute effects; however, recurrent disease was ultimately detected outside of the treatment field and confirmed by biopsy. While BT provided in-field local control, APR was ultimately pursued for potentially curative treatment.

## 3. Treatment Planning and Image-Guidance

Historically, planning in BT has been based on standard loading for a given source and applicator geometry, with two-dimensional plans which resulted in broad generalizability but limited personalization. Since the first utilization of MRI-guided dose–volume adaptation nearly three decades ago [13], and two decades after the Groupe Européen de Curiethérapie (GEC) and European SocieTy for Radiotherapy & Oncology (ESTRO) recommendations endorsed its use for 3D planning [14], image-guided adaptive brachytherapy (IGABT) or four-dimensional BT has become the gold standard for pelvic BT [15].

The use of MRI-guided 3D planning is associated with superior local control rates and reduced toxicity compared to conventional CT-based BT [16] and has the potential to decrease health care costs through a reduction in costs associated with cancer recurrence and treatment toxicities [17]. Despite this, recent survey data indicate that CT-based planning remains the most widely utilized method in clinical practice, with advantages in accessibility, acquisition time, and electronic voxel information [15,18].

Alternate imaging modalities such as US in combination with MRI [19] or both CT/MRI have been explored with favorable outcomes, with one study noting soft tissue target delineation to be superior to CT and comparable to MRI, with considerable advantages in cost and portability [20]. PET/CT-based IGABT is an emerging modality, with 5-year local control rates of 89% for cervical cancer being consistent with MRI-based planning outcomes [15,21].

Below, we describe a unique clinical scenario of a patient with prostate cancer and surgically absent anorectum after abdominoperineal resection (APR) for low rectal cancer, who underwent fluoroscopically guided interstitial BT applicator placement.

### Patient 3

*Clinical Scenario and Indications:* An 87-year-old male presented with elevated PSA and a history of low rectal cancer initially managed with neoadjuvant chemoradiation followed by APR. Surgically absent rectum is typically considered an absolute contraindication for prostate BT and precluded the use of a standard TRUS biopsy. Therefore, diagnosis of prostate cancer was confirmed with CT-guided biopsy, supported by imaging findings and rising PSA. Given his prior pelvic radiation, a cumulative dose to normal tissues from conventional external beam radiation was deemed high risk for treatment-related toxicity, making BT a preferred option.

*Brachytherapy Technique and Applicators:* Image guidance during the procedure was performed using fluoroscopy (Figure 3A) rather than the standard TRUS approach. HDR BT was successfully performed in two sessions using 17 transperineal interstitial needles (Figure 3B), delivering a total of 2700 cGy.

*Outcomes:* The patient’s PSA levels dropped from 9.3 ng/mL to 0.105 ng/mL post treatment, indicating excellent response to therapy. He noted slight transient elevation of lower urinary symptoms from baseline; no high-grade acute or late genitourinary or gastrointestinal toxicity was observed.

## 4. Target Volume Delineation, Dose, and Fractionation

A key consideration in treatment planning is the optimal treatment volume, dose, and fractionation. Various studies have shown dose escalation of relatively radioresistant histologies, such as prostate and gynecologic malignancies, result in improved outcomes [22,23,24], with imaging response to initial chemoradiation helping to further optimize dose [25,26]. While EBRT can be used for dose escalation, the higher integral dose associated with this approach often places OARs at risk of injury. BT is ideally suited for this purpose, so much so that multiple society guidelines identify BT as a vital component of treatment, either as monotherapy (favorable intermediate risk) or boost (high-risk disease) in prostate cancer [27], or as a crucial component of definitive treatment which should not be substituted by EBRT for locally advanced cervical cancer [28].

While pelvic malignancies routinely utilize BT for dose escalation likely due to the greater comfort of providers with the technical aspects of pelvic BT, there are numerous other anatomical sites where BT offers desirable dose escalation while limiting toxicity. Head and neck, gastrointestinal, and soft tissue sarcoma represent a few such malignancies, especially in cases of residual and recurrent disease where increasing data supports broader implementation of BT techniques. A recently published prospective trial examined xerostomia at 6 months in patients with oropharyngeal squamous cell carcinoma who underwent either standard IMRT to 7000 cGy in 35 fractions or IMRT to 5000 cGy in 25 fractions followed by a high-dose rate interstitial BT boost. There was a 30% difference in quantitative xerostomia without any decrement of local control [29]. Similarly, in anal cancer, BT boost using an HDR technique was recently shown to have 5-year colostomy-free-survival rates approaching 90%, which are superior to outcomes reported from EBRT series with low long-term toxicity rates [30]. Consensus guidelines also support the utilization of BT in soft tissue sarcomas, even recommending BT alone in unique clinical scenarios such as the treatment of re-irradiation and pediatric populations, where integral dose considerations are especially important [31].

Another distinct advantage of BT is the availability of several dose/fractionation schedules that produce similar oncologic outcomes and toxicity profiles [32,33]. Such flexibility is especially important in resource-poor environments where a shorter treatment course translates into a decreased burden on patients and institutions. Tata Memorial Hospital reported their institutional experience with an accelerated BT regimen for cervical cancer during the COVID-19 pandemic, utilizing a single insertion and multiple fractions (3–5) over 24–48 h, demonstrating 2-year local control of 88% and low toxicity rates [34]. Similar outcomes have been shown using three fractions delivered during a 24-hour period in a small Australian cohort [35], and in very locally advanced cases in a US safety net health system, delivering four fractions over a 72 h period with a single interstitial implant and adaptive planning [36].

In the clinical case below, we show how oral cavity interstitial BT was utilized to provide an ablative dose to a traditionally radioresistant head and neck cancer subsite with excellent oncologic and quality-of-life outcomes.

### Patient 4

*Clinical Scenario and Indications:* A 68-year-old male with second recurrence of squamous cell carcinoma of the tongue presented for local therapy after refusing any additional surgical procedures. Oral tongue BT is rarely performed due to the anatomical challenges and risks associated with accessing and treating the tongue. However, it was pursued due to the need for a high radiation dose to achieve local control while limiting toxicity.

*Brachytherapy Technique and Applicators:* The patient underwent EBRT using simultaneous integrated boost, with a total of 5000 cGy to the oral tongue, 5625 cGy to the high-risk neck, and 6250 cGy to grossly involved nodes in 25 fractions. Interstitial BT was performed using five catheters placed in two layers ~1 cm apart via a submental approach (Figure 4A), guided by ultrasound to avoid vascular structures. A total dose of 2000 cGy in 5 fractions was delivered as a boost to the oral tongue. Prophylactic tracheostomy was performed prior for airway protection during treatment (Figure 4B).

*Outcomes:* The patient showed an excellent clinical response with near-resolution of the tongue lesion and no residual disease on exam or imaging. At 4 months follow, up he was tolerating an oral diet without dysphagia or odynophagia. Thickened secretions and dysgeusia were the primary toxicities noted and are expected to show ongoing improvement.

## 5. Anesthesia, Analgesia, and Antibiotics

Effective pain management is crucial for optimizing patient outcomes and comfort in BT. The physiological mechanisms of pain during BT are multifaceted and involve both somatic and visceral pathways. Pain from intracavitary BT, particularly for gynecological cancers, is primarily mediated through the sympathetic autonomic afferents entering the spinal cord at the T10-L1 levels, leading to poorly localized, cramping abdominal pain. Additionally, the distension of the cervix and upper vagina stimulates parasympathetic autonomic afferents from the pelvic splanchnic nerves (S2–S4), which can cause lower back pain. Vaginal packing used to stabilize applicators can further exacerbate pain by stimulating somatic afferents via the pudendal nerves, also originating from the S2–S4 spinal roots. This pain can be compounded by the insertion and manipulation of interstitial needles, which may irritate peripheral nerves, causing sharp, localized discomfort [37]. The combination of these pain sources—visceral from deep organ pressure and distension, and somatic from direct nerve irritation—creates a complex pain profile that must be managed effectively to ensure patient comfort and prevent movement during treatment, which could compromise the accuracy of radiation delivery.

A range of analgesic strategies are required to address the diverse pain mechanisms in BT. The choice of analgesia depends on the procedure’s duration, the patient’s pain tolerance, and specific clinical circumstances. Local anesthetics, such as bupivacaine and longer-lasting liposomal bupivacaine, are frequently used for their effective pain control and may reduce post-implant opioid requirements [38]. Interestingly, their effects can last significantly longer than expected; in one example, a single caudal injection for patients undergoing interstitial BT for anal cancer was found to reduce pain for up to 2–3 days, far exceeding the expected duration of the block [37]. Catheter-based techniques, such as continuous epidural infusion, can provide sustained analgesia with lower total doses of anesthetics, minimizing potential side effects while ensuring patient comfort during prolonged BT sessions [39]. General anesthesia may be required for more complex cases wherein complete immobility and pain-free conditions are essential. Ultimately, the choice of anesthesia and analgesia should be tailored to each patient’s needs, optimizing pain relief while minimizing side effects and ensuring a smooth recovery.

The data on the role of periprocedural antibiotics in BT are limited. One study evaluating rates of urinary tract infection (UTI) after prostate BT and cystoscopy found a single perioperative dose of antibiotics resulted in a less than 1% rate of UTI without any postoperative antibiotic use [40]. A larger series evaluating epididymitis after prostate BT, where only half of men received perioperative antibiotics, similarly found rates of 1.2%; higher incidence was noted in men with worse baseline urinary function, suggesting a possible retrograde infection route related to urethral obstruction. These rates are comparable to the 1.0% associated with prostate biopsy and 0.6% with transurethral resection of prostate (TURP), where antibiotic use is nearly universal, raising the question of whether antibiotics truly offer meaningful prophylaxis [41]. For combined intracavitary and interstitial (hybrid) gynecologic BT, post-procedural fevers were noted in 2.4% and 4.9% of patients with and without prophylactic antibiotics, although this difference was not significant [42]. Interestingly, the addition of a non-absorbable antibiotic (rifaximin) in pre-procedural bowel prep was found to reduce rectal volumes, albeit without significant impact on rectal dose–volume histogram (DVH) for intracavitary BT planning [43]. Current evidence does not support routine postoperative antibiotic use; however, individual patient and procedural risks of infection warrant clinical judgment and personalized decision making regarding antibiotic therapy.

## 6. Integration of Genomic Tests and Molecular Markers

Numerous commercially available tests utilizing genomic data have shifted cancer risk stratification from relying solely on clinicopathologic features to genetic and molecular risk assessment. These tests largely help select the optimal treatment pathway on a broad level (e.g., benefit of chemotherapy in breast cancer or androgen deprivation therapy in prostate cancer) but may be especially valuable in malignancies such as uveal melanoma where nearly half of all patients eventually succumb to metastases, despite only 2–4% being identified at presentation. Tools such as DecisionDx-UM [44] can stratify patients such that low-risk, small- to medium-sized tumors may be treated using eye plaque BT, which has shown increasing utilization and superior outcomes to enucleation [45,46], whereas high-risk lesions may benefit from adjuvant therapies or clinical trials. Currently, there are no validated, specific blood or tumor biomarkers for specific use in the context of BT [5,47]. However, existing biomarkers can help with patient selection for BT in two distinct ways.

Firstly, tumors anticipated to show greater resistance to radiation therapy are more likely to show greater response to BT given the higher biologically effective dose (BED) that can be safely achieved. This philosophy is implicit in the commonly utilized salvage BT for pelvic recurrences of prostate cancer, where even hemi-gland BT approaches have shown favorable outcomes [48]. A recent review highlighted nine molecular biomarkers of radiation response shared among the most common malignancies—correlated with DNA repair, signal transduction, hypoxia, and angiogenesis within these tumors—and found vascular endothelial growth factor (VEGF), osteopontin (OPN), and phosphorylated AKT (pAKT) to be the most common radioresistance markers [49]. Inexpensive and clinically accessible laboratory values such as neutrophil count (>7500/μL) [50] and hemoglobin values (<11 g/dL) [51] have also been shown to be markers of radioresistance in cervical cancer patients. Within BT cohorts specifically, the ubiquitous screening test prostate specific antigen (PSA) has shown predictive power on the day of implant (*p* = 0.033) and at one-year follow up (*p* = 0.025) for biochemical failure in high-risk patients undergoing HDR-BT boost after EBRT, and PSA density was predictive even at diagnosis (*p* = 0.009) [52]. Such testing may help identify patient cohorts who may even benefit from dose escalation of BT or addition of radiosensitizers to enhance treatment response.

Conversely, patients who are likely to suffer greater sensitivity to low-dose radiation from EBRT are especially likely to benefit from the tissue-sparing advantage of BT. Unique clinical contexts such as re-irradiation or salvage treatment have extensive published literature on BT utilization, but the advantages extend to specific populations such as children and adolescents, where malignancies such as soft tissue sarcoma and clear cell vaginal cancer are uniquely positioned to benefit from BT given the intrinsic radioresistance of these histologies and greater concern about the late effects of RT [53]. Individuals with germline mutations that increase DNA instability are similarly predisposed to greater toxicity and secondary malignancy from ionizing radiation, such that it is contraindicated in some cases; while specific groups such as those with Fanconi anemia have been treated using EBRT [54], such populations would uniquely benefit from highly conformal BT dose distributions.

Efforts to incorporate radiation sensitivity into RT planning for the broader population have existed for decades. More recently, standardized metrics such as radiation sensitivity index (RSI) [55] and genomic adjusted radiation dose (GARD) are being used not only for prognostication of response [56] but also to characterize distinct phenotypes that might respond to different dose levels from EBRT [57]. Radiosensitivity to BT specifically was recently shown to correlate with markers of DNA double-strand breaks such as γH2AX/53BP1foci at 24 h post LDR prostate seed implant relative to baseline, suggesting subpopulations that may benefit from unique dose, volume, or technique adaptations that further optimize therapeutic ratio [58].

## 7. Artificial Intelligence and Machine Learning

Artificial intelligence (AI) has inherent advantages in personalizing BT planning and a growing body of literature supporting its use; in some cases, output quality is on par with humans, and time and effort are significantly reduced. Most published data focus on prostate and cervical cancer and, regarding EBRT, focus on segmentation tasks. In BT applications, however, the value proposition and complexity are increased given the individualized applicator and needle placement, variety of imaging modalities used (MRI, US), need for segmentation and digitization of seeds or needles, and the importance of minimizing time from implant to initiating treatment. Multiple reviews have found that many auto-segmentation models performed on par or better than human observers [59,60], and time savings could be substantial, with one study showing a model segmentation time of less than 1.5 s compared to up to 20 min for manual segmentation of structures on TRUS [61]. Accuracy for needle segmentation is likewise high, with a less than 1 mm needle tip localization offset reported for CT-based segmentation [62].

Work in image processing and registration has focused on reducing artifact, minimizing distortion across imaging modalities, and improving image ultrasound image quality, with volume overlap metrics exceeding 0.9 in multiple studies [59]. In treatment planning, AI models have been developed for dose calculation and plan optimization. Deep learning models have achieved dosimetric accuracy within 1–2% error compared to conventional methods while offering significant speed improvements [63,64]. Reinforcement learning approaches have shown promise in automating the treatment planning process, with one study reporting plans generated by the deep learning model scoring 10.7% higher in quality score compared to human-generated plans for cervical cancer HDR BT [65].

AI tools have even been explored in BT plan quality assurance. Fan et al. developed a deep neural network for independent verification of dwell positions and times, achieving average deviations of one pixel (~0.5 mm) for positions and within 2% for dwell times [66]. AI models have also been applied to outcome prediction, with studies using convolutional neural networks to predict rectal toxicity in cervical cancer brachytherapy based on dose distribution, finding the upper rectum to be at higher risk of toxicity, similar to data in prostate cancer patients [67].

Despite these promising initial studies, challenges remain in clinical implementation of AI in BT practice. Data quantity and quality are particular concerns given the highly customized nature of BT practices across institutions. Model robustness and generalizability to different clinical scenarios also require further investigation. As the field advances, there is a need for standardization of data collection and reporting, as well as more rigorous clinical validation studies to demonstrate the real-world impact of AI tools in BT practice.

## 8. Challenges and Future Directions

Intensity-modulated brachytherapy (IMBT) represents a significant advancement in personalized brachytherapy, offering improved dose conformity and organ-at-risk sparing. IMBT techniques utilize dynamic source movement or modulated dwell times to optimize dose distributions. One such approach is dynamic modulated brachytherapy (DMBT), which uses a tandem applicator with multiple channels cut into a paramagnetic tungsten alloy rod, allowing for direction modulation of radiation emission through controlled dwell times along each channel. This design has been successfully modeled in commercial treatment planning systems and evaluated with various radioisotopes, including 192Ir, 169Yb, and 60Co, and has shown superior dosimetric profiles compared to conventional intracavitary–interstitial techniques, with potential for reducing the number of interstitial needles required [68]. Challenges remain in the mechanical implementation of these techniques, particularly for interstitial applications. Future research should focus on optimizing source configurations and treatment planning algorithms to fully realize the potential of IMBT across various treatment sites. Additionally, the exploration of novel radionuclides, such as Ytterbium-169, may provide advantages in terms of specific activity and dose distribution for IMBT applications [69].

Shielded BT applicators offer another avenue for enhancing dose distributions in personalized treatments. These approaches use static or dynamic shielding to modulate radiation intensity and improve organ-at-risk sparing. The intracavitary mold applicator (ICMA) for rectal cancer demonstrates the potential of static shielding, with one study showing mean dose to tissues contralateral to the tumor reduced by 24% compared to treatments without shielding [70]. Rotating shield brachytherapy (RSBT) is a more advanced implementation of shielded brachytherapy allowing for dynamic modulation of the radiation beam. Various RSBT approaches have been proposed, including single-shield, dynamic, and paddle-based designs. Yang et al. demonstrated that single-shield RSBT could produce superior high-risk clinical target volume coverage compared to conventional intracavitary and intracavitary–interstitial techniques [71]. The multi-Helix RSBT (H-RSBT) design proposed by Dadkhah et al. represents a promising direction for practical clinical implementation of RSBT with shorter treatment delivery times in cervical cancer patients [72]. In prostate cancer, interstitial RSBT approaches using lower-energy sources like Gadolinium-153 or Ytterbium-169 show potential for urethral sparing or dose escalation, with one study finding urethral sparing of 29–44% compared to conventional HDR brachytherapy [73]. However, challenges in treatment delivery time (154 min vs. 12 min for conventional HDR BT in aforementioned study), shield miniaturization, source production, and delivery apparatus design must be addressed to make these shielded BT approaches clinically feasible. Ongoing developments in 3D printing and advanced materials may facilitate the production of more sophisticated shielded applicators, potentially enabling the creation of patient-specific designs that optimize dose distributions based on individual anatomy and tumor characteristics.

The synergy between BT and the immune system presents an exciting frontier in personalized cancer treatment. One such proposed interaction is radiation-induced in situ tumor vaccination, where radiation converts a patient’s tumor into a nidus for enhanced presentation of tumor-specific antigens [74]. The high doses near the source can maximize immunogenic tumor cell death and antigen release, while intermediate doses may optimally induce phenotypic changes in surviving tumor cells that enhance their susceptibility to immune attack. Lower doses at the tumor periphery may temporarily deplete immunosuppressive cell populations, creating a window of opportunity for enhanced antitumor immune responses. BT’s unique dose distribution, characterized by steep dose gradients and heterogeneous intratumoral doses, may be ideally suited to elicit this effect. 

Other preclinical studies have demonstrated the potential for BT to synergize with immunotherapies. Hodge et al. showed that combining Iodine-125 brachytherapy with a CEA-directed vaccine resulted in reduced lung metastases in a mouse model, demonstrating an abscopal effect [75]. Another study demonstrated that combining HDR BT with dual checkpoint blockade (anti-PD-1 and anti-CD137) led to improved local and distant tumor control in a colorectal cancer model [76]. Future studies should focus on optimizing dose and fractionation schedules to maximize immunogenic effects, as well as exploring combinations with various immunotherapeutic agents. The development of novel brachytherapy applicators capable of delivering both radiation and immunomodulatory drugs could further enhance this synergistic approach. Additionally, investigating the impact of different radionuclides and dose rates on immune responses may provide insights into optimizing BT–immunotherapy combinations. Prospective clinical trials integrating immunotherapy and BT approaches are lacking [74] and are necessary to further advance this promising field.

Expanding the use of BT to less common applications and increasing training programs are crucial for the continued advancement of personalized BT. Recent research has explored robotic BT techniques for challenging sites such as muscle-invasive bladder cancer, demonstrating feasibility in cadaveric models [77]. This approach could potentially offer organ preservation options for patients who are poor candidates for radical cystectomy. However, challenges in applicator design, target delineation, and treatment planning for mobile, hollow organs must be addressed. To support the growth of brachytherapy expertise, initiatives like the American Brachytherapy Society’s 300 in 10 fellowship program and initiatives by ESTRO [78] aim to address the shortage of trained brachytherapists [79]. The development of virtual and augmented reality training platforms, as well as remote planning assistance and telemedicine options, could help democratize access to high-quality BT treatments in underserved regions.

## 9. Conclusions

Personalized BT represents a cutting-edge approach to radiation therapy that underscores the critical importance of tailoring treatment to the unique characteristics and needs of each patient. Over the course of this paper, we have explored the rationale for personalization, technical advancements, and the emerging role of biomarkers, artificial intelligence, and intensity modulation in dose optimization in BT.

As we look to the future, it is evident that personalized BT is poised to play an increasingly vital role in the realm of oncologic treatment. Novel techniques utilizing shielded applicators to improve dose distribution, and immunotherapy in conjunction with BT offer the potential of improve dose conformality and local control. To ensure these advances are widely accessible, the field must maintain efforts to produce well-trained BT practitioners and explore new indications and applications. A personalized approach empowers radiation oncologists to optimize the therapeutic effect while minimizing side effects, enhancing both treatment efficacy and the quality of life for patients.

## Figures and Tables

**Figure 1 cancers-16-03424-f001:**
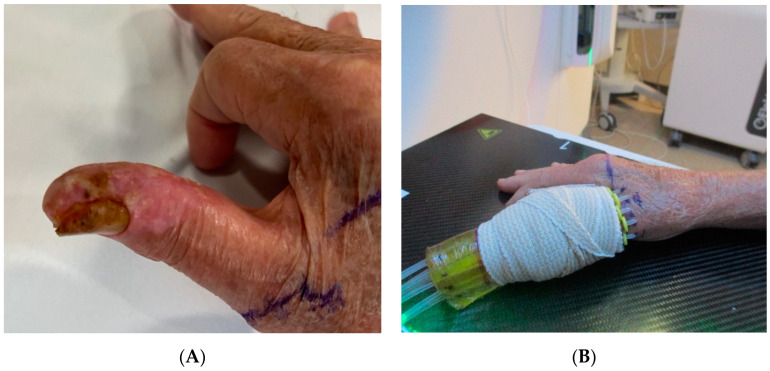
(**A**) Right thumb squamous cell carcinoma prior to radiation therapy. (**B**) Clinical setup utilizing custom surface mold applicators wrapped in elastic bandage.

**Figure 2 cancers-16-03424-f002:**
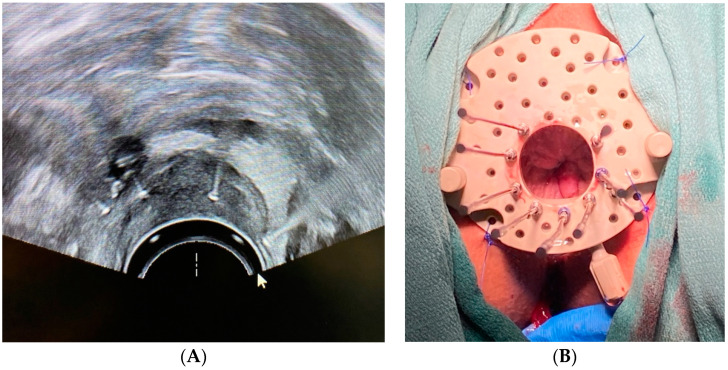
(**A**) Transrectal ultrasound visualization of rectal tumor after needle insertion. (**B**) Kelowna gynecologic template with transperineal needles in place.

**Figure 3 cancers-16-03424-f003:**
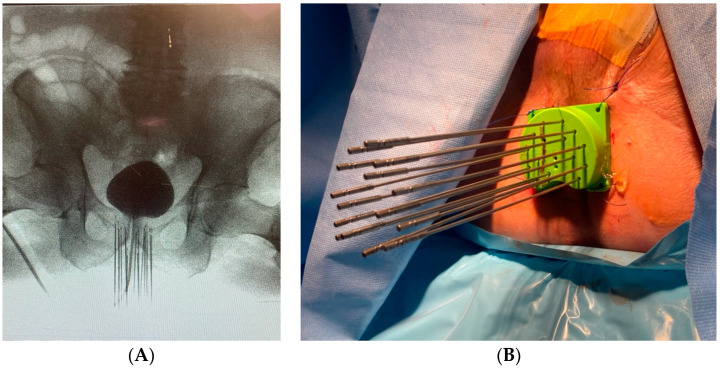
(**A**) Fluoroscopic appearance after transperineal needle insertion with radiopaque contrast in bladder. (**B**) Transperineal needles and template after completion of procedure.

**Figure 4 cancers-16-03424-f004:**
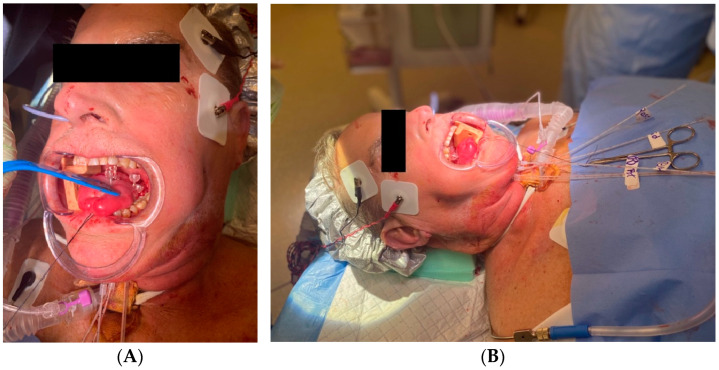
(**A**) Close-up view of oral cavity and interstitial applicators, illustrating submental approach. (**B**) Overall view of patient positioning during procedure, including prophylactic tracheostomy.

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
