# Peer review of "Personalized Brachytherapy: Applications and Future Directions"

_cancers, 2024, doi:10.3390/cancers16193424_

Round 1

Reviewer 1 Report

Comments and Suggestions for Authors

This manuscript delves into the multifaceted realm of personalized brachytherapy, meticulously examining its underlying rationale, diverse applications across medical specialties, recent technological advancements that have revolutionized its practice, and visionary projections outlining its future directions. It will be a solid contribution to the cancers and will certainly appeal to many of its readers, I address some of the main issues with the manuscript in the next few paragraphs. It is recommended that this manuscript be published in cancers after completing revision.

1. The introduction section is too short and should fully discuss the definition of cancer and the limitations of traditional treatment methods. Brachytherapy (BT) and tissue sparing radiotherapy (RT) are also needy to be introduced, including their definition, application scenarios, application scope, advantages, and current challenges.

2. On line 72, "Several studies have shown an overall favorable impact of 3D-printed customized applicators on dosimetric outcomes, repeatability, and procedure time using both low-dose rate (LDR) and high-dose rate (HDR) modalities, including some prospective randomized data" is mentioned. What is the favorable impact? Prospective randomized data should be listed.

3. When describing Patient 1, mention that she is immunosuppressed. So what is causing the immunosuppression in this patient?

4. When describing the patient's condition, some necessary test data should be cited to demonstrate the severity of the patient. Evidence of improvement in the patient's condition should also be included. 

5. With too few cases, it is impossible to fully prove the effectiveness of RT and BT.

6. This manuscript illustrates thatVarious studies have shown dose escalation of relatively radioresistant histologies such as prostate and gynecologic malignancies result in improved outcomes, with imaging response to initial chemoradiation helping to further optimize dose.” To support this statement, the following recently published important related papers should be cited: Chem. Soc. Rev. 2021, 50, 2839; Adv Mater. 2022, 34, 2106388; Adv. Mater. 2024, 36, 2304249; Cancer Drug Resist 2023, 6, 805; Exploration 2023, 3, 20210111; VIEW. 2023, 4, 20220064; Coord. Chem. Rev. 2024, 517, 216054.

Comments on the Quality of English Language

Extensive editing of English language required.

Author Response

Reviewer 1 Comments and Responses:

We thank the reviewer for taking the time to review and provide commentary on our review article. Below are our responses to the provided suggestions.

  1. The introduction section is too short and should fully discuss the definition of cancer and the limitations of traditional treatment methods. Brachytherapy (BT) and tissue sparing radiotherapy (RT) are also needy to be introduced, including their definition, application scenarios, application scope, advantages, and current challenges.

This review article is intended for clinicians who are familiar with basics of radiotherapy and definitions of BT and RT. The remainder of the review article focuses on the applications, advantages, and challenges so repeating that in the introduction would be redundant.

  1. On line 72, "Several studies have shown an overall favorable impact of 3D-printed customized applicators on dosimetric outcomes, repeatability, and procedure time using both low-dose rate (LDR) and high-dose rate (HDR) modalities, including some prospective randomized data" is mentioned. What is the favorable impact? Prospective randomized data should be listed.

The prospective randomized paper is now mentioned by author name and supporting references are included.

  1. When describing Patient 1, mention that she is immunosuppressed. So what is causing the immunosuppression in this patient?

The patient scenario has been updated to reflect that this is due to prior cardiac transplant.

  1. When describing the patient's condition, some necessary test data should be cited to demonstrate the severity of the patient. Evidence of improvement in the patient's condition should also be included.  

Including excessive amounts of clinical data would add to the length without meaningfully adding to the review article. This is not meant to be a case report or case series, but rather as a brief example of possible unique applications of brachytherapy.

  1. With too few cases, it is impossible to fully prove the effectiveness of RT and BT.

The clinical cases are for illustrative purposes as examples for clinicians to understand possible application strategies. The effectiveness of RT and BT is shown by the numerous studies that are referenced in the paper and many others.

  1. This manuscript illustrates that“Various studies have shown dose escalation of relatively radioresistant histologies such as prostate and gynecologic malignancies result in improved outcomes, with imaging response to initial chemoradiation helping to further optimize dose.” To support this statement, the following recently published important related papers should be cited:  Soc. Rev.202150, 2839; Adv Mater. 202234, 2106388; Adv. Mater. 202436, 2304249; Cancer Drug Resist 20236, 805; Exploration 2023, 3, 20210111; VIEW. 2023, 4, 20220064; Coord. Chem. Rev. 2024, 517, 216054.

It is not clear what papers the reviewer is referencing. The first paper is a paper titled “Supramolecular cancer nanotheranostics” with no relevance to brachytherapy and the remaining do not show up in pub med. If the reviewer is able to provide some PMID’s or at least paper titles we can consider including those references, however we already include 3 references which we feel should be adequate to support this statement.

Reviewer 2 Report

Comments and Suggestions for Authors The text discussed is an alaysis on personalized brachytherapy, exploring its applications, technological innovations such as 3D-printed applicators, and the potential of artificial intelligence (AI). Despite progress, challenges include the need for more training and personalized treatments to improve cancer care effectiveness. The topic is well investigated and described, and in my opinion, apart from a minor revision of the English language, it does not require further changes to be accepted.

Author Response

Reviewer 2 Comments and Responses:

We thank the reviewer for taking the time to review and provide commentary on our review article. Below are our responses to the provided suggestions.

  1. The text discussed is an alaysis on personalized brachytherapy, exploring its applications, technological innovations such as 3D-printed applicators, and the potential of artificial intelligence (AI). Despite progress, challenges include the need for more training and personalized treatments to improve cancer care effectiveness. The topic is well investigated and described, and in my opinion, apart from a minor revision of the English language, it does not require further changes to be accepted.

We appreciate the kind feedback. We will ensure authors review the manuscript for any minor language revisions that may be needed.

Reviewer 3 Report

Comments and Suggestions for Authors

Comments to the authors:

I have reviewed the manuscript, entitled "Personalized Brachytherapy: Applications and Future Directions." I have some detailed comments for improvement and polishing:

1: The title is clear but could be more focused. Consider emphasizing the future directions and applications of personalized brachytherapy to provide more clarity on the paper's aim, such as "Advancements and Future Directions in Personalized Brachytherapy: Techniques and Applications." Further, the abstract is well-written and gives a solid overview. However, you could refine the following: You mention "advanced imaging techniques" and "technological advances." It would help the reader if you list the specific techniques like MRI or PET/CT early on. While you mention challenges such as training practitioners, it's also useful to suggest potential solutions (e.g., virtual simulations, more fellowships).

2: The list of brachytherapy advantages is clear but could be made more engaging by linking these points to specific patient outcomes. "BT’s steep dose gradient, proximity to the tumor, and adaptability make it exceptionally suited to minimize toxicities while improving local control in various cancer types, such as prostate and cervical cancers."

3: The section on 3D-printed applicators is informative but dense. Consider breaking it into smaller paragraphs, each addressing specific aspects—clinical feasibility, cost, dosimetric outcomes, etc.  Example: “The advent of 3D printing has revolutionized BT, particularly for complex cases like gynecological cancers. A recent review reports that gynecological applications represent 32% of all cases utilizing 3D-printed templates, followed by skin (19%) and head and neck (9%). These developments have led to improvements in precision and cost reduction." The case studies are engaging but lack depth in terms of outcome analysis. You might include follow-up details, such as recurrence rates, survival, or quality of life metrics, where relevant. Adding supporting literature (even speculative references) can be beneficial.

4: When discussing MRI, PET/CT, and ultrasound-guided planning, there’s room to expand on specific advantages and disadvantages of each modality in more detail. For example: “MRI-guided 3D planning, while associated with superior local control, is often limited by accessibility and acquisition time, whereas PET/CT may offer a cost-effective alternative without compromising treatment outcomes in cervical cancer." Moreover, the discussion of AI’s role in BT is promising, but more depth is needed in areas such as clinical implementation or existing challenges, especially concerning data standardization across institutions.

5: The section on combining BT with immunotherapy is a strong point, but it is briefly covered. Consider expanding on specific preclinical or early clinical trials, providing more insight into potential immunotherapy combinations.

6: The conclusion could focus more on future directions and challenges in BT, such as enhancing shielded applicators or improving real-time AI applications.

7:

Comments on the Quality of English Language

Minor editing of English language required.

Author Response

Reviewer 3 Comments and Responses:

We thank the reviewer for taking the time to review and provide commentary on our review article. We overall acknowledge the possible benefits of many of the specific commentaries provided below but as a general theme it seems the reviewer prefers a more thorough and detailed approach to each topic, whereas the philosophical goal of our approach to this paper is to provide an overview of many of the topics and give clinicians and understanding of the possible novel approaches that can be used in brachytherapy and upcoming new directions. At that the current length of our manuscript, we feel it would compromise the readability of the paper and make it unbalanced to take a deeper dive into specific subsections and leave the more thorough evaluation of topics such as “BT+3D printing” or “BT+AI” to either review articles or studies focused on these topics alone. Below are our responses to the provided suggestions.

  1. 1: The title is clear but could be more focused. Consider emphasizing the future directions and applications of personalized brachytherapy to provide more clarity on the paper's aim, such as "Advancements and Future Directions in Personalized Brachytherapy: Techniques and Applications." Further, the abstract is well-written and gives a solid overview. However, you could refine the following: You mention "advanced imaging techniques" and "technological advances." It would help the reader if you list the specific techniques like MRI or PET/CT early on. While you mention challenges such as training practitioners, it's also useful to suggest potential solutions (e.g., virtual simulations, more fellowships).

We appreciate title suggestions but favor a more concise title and feel this conveys the subject matter well. In the abstract we aim to stay more general and define the specific imaging and technological advances in the corresponding sections of the body of the manuscript.

  1. 2: The list of brachytherapy advantages is clear but could be made more engaging by linking these points to specific patient outcomes. "BT’s steep dose gradient, proximity to the tumor, and adaptability make it exceptionally suited to minimize toxicities while improving local control in various cancer types, such as prostate and cervical cancers."

The specific advantage in these cancer types is mentioned later in the manuscript “Various studies have shown dose escalation of relatively radioresistant histologies such as prostate and gynecologic malignancies result in improved outcomes … BT is ideally suited for this purpose”. We aim to keep the introduction portion more general while providing specific examples as appropriate in the body of the text.

  1. 3: The section on 3D-printed applicators is informative but dense. Consider breaking it into smaller paragraphs, each addressing specific aspects—clinical feasibility, cost, dosimetric outcomes, etc.  Example: “The advent of 3D printing has revolutionized BT, particularly for complex cases like gynecological cancers. A recent review reports that gynecological applications represent 32% of all cases utilizing 3D-printed templates, followed by skin (19%) and head and neck (9%). These developments have led to improvements in precision and cost reduction." The case studies are engaging but lack depth in terms of outcome analysis. You might include follow-up details, such as recurrence rates, survival, or quality of life metrics, where relevant. Adding supporting literature (even speculative references) can be beneficial.

Given that there are entire systematic reviews of 3D printed applicators in brachytherapy (Fahimian et al, Brachytherapy 2023) and that it is only one approach to equipment customization, we felt that adding additional depth would unbalance the paper – particularly since we show cases of repurposing existing templates and imaging techniques such as fluoroscopy which fall outside the scope of 3D printing. As far as adding more depth to the cases, we actually initially had more detailed case reports but the manuscript length became significantly longer without relevance to the subject at hand which is the practical implementation of a brachytherapy technique. We thus chose to cut down on this extra information in favor of greater readability.

  1. 4: When discussing MRI, PET/CT, and ultrasound-guided planning, there’s room to expand on specific advantages and disadvantages of each modality in more detail. For example: “MRI-guided 3D planning, while associated with superior local control, is often limited by accessibility and acquisition time, whereas PET/CT may offer a cost-effective alternative without compromising treatment outcomes in cervical cancer." Moreover, the discussion of AI’s role in BT is promising, but more depth is needed in areas such as clinical implementation or existing challenges, especially concerning data standardization across institutions.

There exist numerous studies and specific review articles discussing imaging in BT and AI and BT as a stand alone subject in greater depth (see Drs. Jia and Albuquerque paper in Seminars in Radiation Oncology 2022). Our goal is to provide a more concise and brief overview of these topics, and readers desiring more comprehensive overviews of these topics can reference the appropriate articles.

  1. 5: The section on combining BT with immunotherapy is a strong point, but it is briefly covered. Consider expanding on specific preclinical or early clinical trials, providing more insight into potential immunotherapy combinations.

This topic has very limited studies in the clinical space as noted in the review article we referenced. We include a few preclinical studies but felt that adding extensive preclinical data would deviate from the goal of a more practical and clinically focused paper. We have modified this section to highlight this lack of clinical data:

“Prospective clinical trials integrating immunotherapeutic and BT approaches are lacking [74], and are necessary to further advance this promising field.”

  1. 6: The conclusion could focus more on future directions and challenges in BT, such as enhancing shielded applicators or improving real-time AI applications.

We have modified the conclusion to address future directions and challenges and agree this makes the conclusion stronger:
“Novel techniques utilizing shielded applicators to improve dose distribution, and immunotherapy in conjunction with BT offer the potential of improve dose conformality and local control. To ensure these advances are widely accessible, the field must maintain efforts to produce well-trained BT practitioners and explore new indications and applications.”

Round 2

Reviewer 3 Report

Comments and Suggestions for Authors

The authors have addressed all the concerns